# Real-Time Detection and Quantitative Analysis of Spurious Forgetting in Continual Learning

## Abstract

Catastrophic forgetting remains a fundamental challenge in continual learning for large language models. Recent work (Zheng et al., 2025) revealed that performance degradation may stem from *spurious forgetting* caused by task alignment disruption rather than true knowledge loss. However, this foundational work left critical gaps: it only qualitatively describes alignment, relies on post-hoc analysis, and lacks automatic distinction mechanisms.

**Key Contribution:** We extend (Zheng et al., 2025) by introducing the *shallow versus deep alignment* framework, which provides the first quantitative characterization of alignment depth. We identify that current task alignment approaches suffer from *shallow alignment*—alignment is maintained only over the first few output tokens (approximately 3-5), making models vulnerable to forgetting. This shallow alignment explains why spurious forgetting occurs, why it is reversible, and why fine-tuning attacks are effective.

In this paper, we propose a comprehensive framework that addresses all gaps in (Zheng et al., 2025): (1) **quantitative metrics** (0-1 scale) to measure alignment depth across token positions, addressing the qualitative-only limitation; (2) **real-time detection methods** for identifying shallow alignment and spurious forgetting during training, enabling early intervention; (3) **specialized analysis tools** for alignment depth visualization and recovery prediction; and (4) **adaptive mitigation strategies** that automatically distinguish forgetting types and promote deep alignment. Extensive experiments on multiple datasets and model architectures (Qwen2.5-3B to Qwen2.5-32B) demonstrate 86.2-90.6% identification accuracy and show that promoting deep alignment improves robustness against forgetting by 3.3-7.1% over baselines, including the fixed freezing strategy in (Zheng et al., 2025).

## 1. Introduction

Continual learning has emerged as a critical capability for large language models (LLMs) to adapt to new tasks and domains without forgetting previously acquired knowledge. As LLMs are increasingly deployed in dynamic environments where new tasks and domains emerge continuously, the ability to learn new capabilities while preserving existing ones becomes essential for practical applications. However, the phenomenon of catastrophic forgetting, where models lose performance on previous tasks when learning new ones, poses a significant challenge (McCloskey & Cohen, 1989). This problem is particularly acute in resource-constrained scenarios where storing all training data for replay is infeasible, or when privacy concerns prevent data retention. Traditional approaches assume that performance degradation directly indicates knowledge loss, leading to strategies that attempt to preserve all learned parameters or replay all previous data. (see Section A.5.1)

Recent research has revealed a more nuanced understanding of forgetting mechanisms. The concept of *spurious forgetting*, introduced in 2025, suggests that performance degradation may stem from task alignment disruption rather than true knowledge loss (Zheng et al., 2025). In spurious forgetting, internal representations remain intact, but the alignment between representations and the output layer is disrupted. This distinction is crucial because spurious forgetting can be reversed through minimal fine-tuning (often requiring only 50-100 samples and 1-3 epochs), whereas true forgetting requires extensive retraining with full datasets. Understanding this distinction opens new opportunities for efficient mitigation: instead of preserving all parameters or replaying all data, we can focus on maintaining or repairing alignment, which is far more computationally efficient.

However, we identify a fundamental limitation in current task alignment approaches: *task alignment is largely only*

[1]Anonymous Institution, Anonymous City, Anonymous Region, Anonymous Country. Correspondence to: Anonymous Author <anon.email@domain.com>.

Preliminary work. Under review by the International Conference on Machine Learning (ICML). Do not distribute.

*a few tokens deep*—what we term *shallow alignment*. This shallow alignment creates critical vulnerabilities that explain why spurious forgetting occurs, why it is reversible, and why fine-tuning attacks are effective. (see Section A.5.2)

**The Shallow Alignment Problem:** Alignment adapts the model's generative distribution primarily over only the very first few output tokens (approximately 3-5 tokens)—*shallow alignment*. This shallow alignment creates a critical vulnerability: if initial tokens deviate from expected alignment (due to new task training, adversarial manipulation, or distribution shift), generation catastrophically falls onto a harmful trajectory of forgetting, even though underlying representations remain intact.

This shallow alignment problem provides a unified explanation for multiple forgetting phenomena: (1) **Spurious forgetting**—alignment disruption in initial tokens leads to apparent performance loss, even when knowledge is preserved; (2) **Reversibility**—since only shallow alignment is affected, recovery is possible with minimal intervention (fine-tuning output layers only); (3) **Fine-tuning vulnerability**—modifying first few tokens can undo alignment, explaining why few fine-tuning steps can lead to forgetting; (4) **Freezing effectiveness**—freezing bottom layers protects representations while allowing shallow alignment to adapt, explaining why this strategy works.

We advocate *deep alignment*, where models maintain alignment consistently across multiple token positions (approximately 10-20 tokens), ensuring robustness even when initial tokens are perturbed. (see Section A.5.3)

**Our main contributions:** Building upon (Zheng et al., 2025)'s foundational work on spurious forgetting, we make four key advances:

(1) **Unified theoretical framework**—introduce shallow versus deep alignment, providing the first quantitative characterization of alignment depth and unified explanation for multiple forgetting phenomena (spurious forgetting, reversibility, fine-tuning vulnerability, freezing effectiveness).

(2) **Quantitative measurement framework**—quantitative framework (0-1 scale) for measuring alignment depth across token positions, addressing the qualitative-only gap in (Zheng et al., 2025) which only describes alignment as "aligned/not aligned".

(3) **Real-time detection system**—comprehensive framework for identifying shallow alignment and spurious forgetting in real-time during training, addressing the post-hoc analysis limitation in (Zheng et al., 2025).

(4) **Deep alignment training and adaptive mitigation**—proactive training strategies that promote deep alignment from the start, and adaptive mitigation strategies that automatically distinguish forgetting types, extending (Zheng

et al., 2025)'s fixed freezing strategy to adaptive, type-specific approaches.

Experimental validation (Qwen2.5-3B to Qwen2.5-32B) demonstrates 86.2-90.6% identification accuracy and 3.3-7.1% improvement over baselines, including (Zheng et al., 2025)'s fixed freezing strategy. (see Section B.2.14)

## 2. Related Work

### 2.1. Catastrophic Forgetting

Early approaches to catastrophic forgetting focused on three main paradigms: (1) **Regularization-based methods**—preserve important parameters through weight constraints (EWC (Kirkpatrick et al., 2017), SI (Zenke et al., 2017)), preventing large changes to parameters identified as important for previous tasks; (2) **Experience replay**—store and replay samples from previous tasks during new task training (Rolnick et al., 2019), maintaining performance through data retention; (3) **Parameter isolation**—allocate separate parameters for different tasks (Mallya & Lazebnik, 2018), avoiding interference through architectural design. While these methods have shown effectiveness in certain scenarios, they often incur significant computational overhead (experience replay can require 30-50% additional computation) or storage costs (parameter isolation can double model size). Recent work explores forgetting in LLMs (Luo et al., 2023), with strategies including hierarchical model merging (Wang et al., 2025) and negative preference optimization (Zhang et al., 2024). However, these approaches treat all performance degradation as true forgetting, missing opportunities to efficiently recover from spurious forgetting. Our work addresses this gap by providing mechanisms to distinguish and handle different forgetting types. A detailed comparison of these approaches is provided in Section B.2.14.

### 2.2. Spurious Forgetting

The concept of spurious forgetting was recently introduced in 2025 (Zheng et al., 2025) (ICLR 2025), showing that task alignment disruption can cause apparent forgetting even when internal representations remain intact. This foundational work made several key contributions that establish the theoretical foundation for our work.

**Key Contributions of (Zheng et al., 2025):**

1. **Conceptual Foundation:** First introduced the concept of spurious forgetting, distinguishing it from true forgetting based on representation preservation. This distinction is crucial because spurious forgetting can be reversed through minimal fine-tuning, whereas true forgetting requires extensive retraining.

2. **Freezing Strategy:** Proposed freezing bottom layers

(approximately 30% of layers) to mitigate spurious forgetting by protecting representations while allowing output layer adaptation. This strategy demonstrates that protecting deep representations can prevent alignment disruption.

3. **Reversibility Demonstration:** Showed that minimal fine-tuning (often just 50-100 samples, 1-3 epochs) can restore performance when spurious forgetting occurs, confirming that knowledge is preserved in the representation space.

4. **Theoretical Analysis:** Connected alignment shifts to orthogonal updates in model weights, providing theoretical foundation for understanding alignment disruption. The work showed that orthogonal updates primarily affect the output layer, explaining why freezing bottom layers is effective.

**Limitations of (Zheng et al., 2025):** While (Zheng et al., 2025) provides important insights, it left four critical gaps that limit practical applicability:

1. **No Quantitative Metrics:** Alignment was only qualitatively described as "aligned/not aligned", without continuous measurement or depth characterization. This prevents precise understanding of alignment strength and depth, making it impossible to measure how deeply alignment is maintained across token positions.

2. **No Real-Time Detection:** Identification relies on post-hoc analysis after forgetting has occurred, missing opportunities for early intervention when alignment becomes shallow. This reactive approach cannot prevent forgetting before it causes performance degradation.

3. **No Automatic Distinction:** Cannot automatically distinguish true from spurious forgetting, requiring manual analysis and expert knowledge. This limits scalability and practical deployment in real-world scenarios.

4. **No Specialized Tools:** Lacks tools for measuring alignment depth, identifying shallow alignment, and predicting recovery requirements. Researchers and practitioners cannot easily analyze alignment dynamics or predict recovery needs.

**Our Extension:** Our work addresses all these gaps by introducing the *shallow versus deep alignment* framework, which: (1) provides quantitative metrics (0-1 scale) to measure alignment depth across token positions, enabling precise characterization; (2) enables real-time detection during training, allowing proactive mitigation; (3) automatically distinguishes forgetting types through integrated scoring;

(4) provides specialized analysis tools (alignment depth analyzer, reversibility analyzer, dynamic tracker). Additionally, we extend (Zheng et al., 2025)'s fixed freezing strategy to adaptive, type-specific mitigation strategies, and introduce deep alignment training to prevent shallow alignment from occurring. This comprehensive extension transforms (Zheng et al., 2025)'s foundational insights into a complete, actionable framework. (see Section B.2.14)

### 2.3. Representation Space Analysis

Understanding how representations change during continual learning is crucial for distinguishing different types of forgetting. CKA (Centered Kernel Alignment) (Kornblith et al., 2019) measures representation similarity between different model states, enabling comparison of internal representations across tasks. PCA (Principal Component Analysis) enables visualization of high-dimensional representations in lower-dimensional spaces, helping researchers understand representation changes. These tools have been widely used to analyze forgetting in neural networks, revealing that representations can remain similar even when performance degrades. However, these approaches focus on representation-level analysis and lack specialized tools for measuring alignment depth and identifying shallow alignment. Our alignment depth metrics extend these approaches by quantifying how deeply alignment is maintained across token positions, not just whether representations are similar. This provides a more nuanced understanding of forgetting mechanisms: even when representations are similar, shallow alignment can cause apparent forgetting, which our metrics can detect and quantify. Detailed discussion is provided in Section B.2.14.

**Summary:** Existing approaches share a common limitation: they treat all performance degradation as true forgetting and lack quantitative, real-time mechanisms to distinguish and handle different forgetting types. Our work addresses these limitations by introducing the shallow versus deep alignment framework with quantitative metrics, real-time detection, and adaptive mitigation strategies. This framework extends (Zheng et al., 2025)'s qualitative understanding to a quantitative, actionable solution.

## 3. Theoretical Framework

### 3.1. Shallow vs Deep Alignment

While (Zheng et al., 2025) identified that task alignment disruption causes spurious forgetting, it only qualitatively describes alignment without measuring alignment depth. This limitation prevents understanding why alignment is vulnerable and how to make it robust. We extend (Zheng et al., 2025) by introducing quantitative metrics to measure alignment depth across token positions, enabling us to dis-

tinguish shallow alignment (vulnerable to disruption) from deep alignment (robust against perturbations). This quantitative characterization reveals a fundamental limitation: current task alignment approaches are largely only a few tokens deep, explaining the vulnerability observed in (Zheng et al., 2025).

**Connection to (Zheng et al., 2025)'s Orthogonal Updates:** (Zheng et al., 2025) connected alignment shifts to orthogonal updates in model weights, showing that these updates primarily affect the output layer. Our shallow versus deep alignment framework provides a quantitative explanation: orthogonal updates during new task training primarily disrupt shallow alignment (the first few tokens), while deep representations remain intact. This explains why (Zheng et al., 2025)'s freezing strategy works—by protecting bottom layers, it prevents disruption of deep representations while allowing shallow alignment to adapt. Our quantitative framework measures alignment depth $D(\theta, \mathcal{T})$, revealing that standard training leads to shallow alignment ($D \leq 3$), while (Zheng et al., 2025)'s freezing maintains this shallow alignment but protects underlying knowledge. Our deep alignment training strategies go further by promoting deep alignment ($D > 12$) from the start, preventing shallow alignment vulnerability.

We first formally characterize the problem. Let $\mathcal{M}$ be a model with parameters $\theta$, and $\mathcal{T}_1, \mathcal{T}_2$ be two tasks. After training on $\mathcal{T}_1$, performance is $P_1(\theta_1)$. When learning $\mathcal{T}_2$, parameters change to $\theta_2$, and $P_1(\theta_2) < P_1(\theta_1)$.

**Shallow Alignment** occurs when task alignment primarily depends on only the first few output tokens. Formally, let $A_t(\theta, \mathcal{T})$ denote the alignment score at token position $t$. Shallow alignment is characterized by:

$$A_t(\theta, \mathcal{T}) \gg A_{t'}(\theta, \mathcal{T}) \quad \text{for } t \leq k, t' > k \qquad (1)$$

where $k$ is a small constant (approximately $k \leq 5$), meaning alignment is strong only in the first few tokens but weak in later tokens. This shallow alignment explains why (Zheng et al., 2025)'s orthogonal updates primarily affect the output layer: the output layer controls alignment for the first few tokens, making it vulnerable to disruption.

**Deep Alignment** occurs when alignment is maintained across multiple token positions:

$$A_t(\theta, \mathcal{T}) \geq \tau_{\text{deep}} \quad \text{for } t \in \{1, 2, \ldots, T\} \qquad (2)$$

where $T$ is the typical sequence length and $\tau_{\text{deep}}$ is a threshold indicating sufficient alignment depth. Deep alignment provides robustness: even if the first few tokens are disrupted (as in (Zheng et al., 2025)'s orthogonal updates), subsequent tokens maintain correct alignment, allowing the model to self-correct.

### 3.2. Why Shallow Alignment Exists

We identify both theoretical and practical reasons why shallow alignment emerges. In transformer architectures, gradient flow exhibits a natural bias toward early tokens due to the attention mechanism, resulting in gradient magnitudes decreasing exponentially with token position ($\|\nabla_\theta \mathcal{L}_t\| \propto \alpha^t$ where $\alpha \approx 0.6\text{-}0.7$). This gradient bias causes optimization to naturally prioritize early tokens, leading to shallow alignment. Standard training procedures (e.g., early stopping, limited epochs) further reinforce this bias by primarily penalizing early token misalignments. (see Section A.1 for detailed theoretical derivation and empirical validation)

### 3.3. Empirical Analysis of Shallow Alignment in Existing Methods

We empirically analyze why existing methods, including (Zheng et al., 2025)'s fixed freezing strategy, maintain shallow alignment. Our analysis reveals that while these methods may protect representations or maintain performance, they do not actively promote deep alignment across multiple token positions.

**Fixed Freezing Strategy:** While (Zheng et al., 2025)'s fixed freezing strategy (freezing bottom 30% layers) protects representations, it does not promote deep alignment. Our measurements on CLINC-150 and 20 Newsgroups show that models trained with fixed freezing achieve alignment depth $D \leq 3$ on average, similar to standard training. This is because fixed freezing only prevents alignment disruption but does not actively promote alignment across multiple token positions. The strategy protects deep representations (maintaining representation similarity $> 0.85$) but allows shallow alignment to remain vulnerable (alignment depth $D \leq 3$). Our quantitative measurements confirm that fixed freezing maintains shallow alignment: for Qwen2.5-3B on CLINC-150, fixed freezing achieves $D = 2.9$ compared to $D = 2.8$ for standard training, demonstrating that freezing alone does not promote deep alignment.

**Standard Training:** Standard training procedures naturally lead to shallow alignment due to gradient flow bias. Our empirical measurements on CLINC-150 and 20 Newsgroups show that standard training achieves $D \leq 3$ on average, with alignment scores dropping below threshold $\tau_{\text{deep}} = 0.7$ after the first 3-5 tokens. Specifically, for Qwen2.5-3B, we observe $A_1 = 0.85$, $A_3 = 0.72$, $A_5 = 0.58$, and $A_{10} = 0.42$, confirming that alignment is strong only for the first few tokens. This empirical finding supports our theoretical analysis of gradient flow bias.

**Experience Replay:** While experience replay helps maintain performance, it does not promote deep alignment. Our measurements show that experience replay achieves $D \leq 4$ on average, only slightly better than standard train-

ing ($D \leq 3$). For Qwen2.5-3B on CLINC-150, experience replay achieves $D = 3.2$ compared to $D = 2.8$ for standard training. This marginal improvement suggests that replaying data helps maintain shallow alignment but does not actively promote deep alignment across multiple token positions.

**Regularization Methods:** Methods like EWC (Elastic Weight Consolidation) also maintain shallow alignment. Our measurements show that EWC achieves $D \leq 3.5$ on average, similar to standard training. This is because regularization methods focus on preserving important parameters but do not actively promote alignment across multiple token positions.

These empirical findings support our theoretical analysis and demonstrate the need for deep alignment training strategies that actively promote alignment across multiple token positions. The key insight is that protecting representations (as in fixed freezing) or maintaining performance (as in experience replay) is not sufficient—we must actively promote deep alignment to achieve robustness against forgetting. (see Section A.1)

### 3.4. Definition of Spurious Forgetting

Given the shallow alignment framework, we can now precisely define forgetting types. This distinction is crucial because different forgetting types require different mitigation strategies: spurious forgetting can be quickly reversed through targeted fine-tuning, while true forgetting requires extensive retraining.

**True Forgetting** occurs when internal representations are fundamentally altered. In true forgetting, the model's internal knowledge about the task is lost, requiring experience replay or extensive retraining to recover. Formally:

$$\|\mathbf{R}_1(\theta_1) - \mathbf{R}_1(\theta_2)\| > \tau_{\text{true}} \tag{3}$$

where $\mathbf{R}_1(\theta)$ represents internal representations for task $\mathcal{T}_1$ (computed as hidden layer activations averaged over task data), and $\tau_{\text{true}} = 0.3$ is a threshold determined through empirical validation. True forgetting often occurs when: (1) new task training uses high learning rates or many epochs, causing large parameter changes; (2) new task data distribution is very different from previous tasks; (3) no preservation mechanisms (freezing, regularization) are applied.

**Spurious Forgetting** occurs when representations remain similar but shallow alignment is disrupted. In spurious forgetting, knowledge is preserved (representations are similar), but the alignment between representations and output layer is broken. Formally:

$$\|\mathbf{R}_1(\theta_1) - \mathbf{R}_1(\theta_2)\| \leq \tau_{\text{true}} \quad \text{and} \quad A_1(\theta_2, \mathcal{T}_1) < \tau_{\text{align}} \tag{4}$$

where $A_1(\theta, \mathcal{T})$ is the alignment score for the first token, and $\tau_{\text{align}} = 0.7$ is the alignment threshold. The key in-

sight is that spurious forgetting is a manifestation of shallow alignment: when only the first few tokens are misaligned (due to output layer changes during new task training), the model appears to forget, even though deeper representations remain intact. This explains why spurious forgetting is reversible with minimal intervention—only the shallow alignment layer (output layer) needs to be repaired, not the entire representation space. (see Section A.2)

### 3.5. Mechanism of Shallow Alignment Leading to Spurious Forgetting

In autoregressive generation, when alignment is shallow ($D \leq 5$), misalignment of initial tokens cascades through subsequent tokens, leading to complete misalignment. When initial tokens are misaligned due to output layer changes during new task training, the misalignment propagates through the generation process, causing apparent forgetting even when deep representations remain intact. Since only shallow alignment is affected, recovery requires minimal intervention—fine-tuning only the output layer can restore alignment with minimal data (50-100 samples) and epochs (1-3 epochs). (see Section A.2 for detailed mathematical characterization)

### 3.6. Quantitative Alignment Metric and Depth Measurement

Unlike previous work (Zheng et al., 2025) which only qualitatively describes alignment, we introduce quantitative metrics to measure alignment depth. We define alignment score $A(\theta, \mathcal{T})$ as a continuous measure $[0, 1]$ computed from hidden representations and output layer weights using cosine similarity. Token-level alignment scores $A_t(\theta, \mathcal{T})$ measure alignment at each position $t$. The alignment depth $D(\theta, \mathcal{T}) = \max\{k : A_t(\theta, \mathcal{T}) \geq \tau_{\text{deep}} \text{ for all } t \leq k\}$ measures how many consecutive tokens maintain sufficient alignment (above threshold $\tau_{\text{deep}} = 0.7$). A model with $D(\theta, \mathcal{T}) \leq 5$ exhibits shallow alignment, while $D(\theta, \mathcal{T}) > 10$ indicates deep alignment. (see Section A.3 for detailed mathematical formulation)

### 3.7. Reversibility and Detection

Detecting spurious forgetting and predicting recovery potential are essential for efficient mitigation. We introduce two key scores: reversibility score $R(\theta, \mathcal{T})$ measures recovery potential by combining alignment score, representation similarity (CKA), and gradient norm. The spurious forgetting score $S(\theta, \mathcal{T})$ combines alignment drop, reversibility, and performance degradation. High $S$ ($> 0.6$) with high $R$ ($> 0.6$) and low $A$ ($< 0.7$) indicates spurious forgetting. Optimal thresholds ($\tau_S = 0.6$, $\tau_R = 0.6$, $\tau_{\text{align}} = 0.7$) are determined through extensive validation, achieving 86.2-90.6% identification accuracy. (see Section A.4 for detailed

threshold selection rationale)

# 4. Methodology

Our framework consists of three main components: (1) real-time detection of shallow alignment, (2) specialized analysis tools for understanding alignment depth, and (3) adaptive mitigation strategies that promote deep alignment. (see Section B.1.1)

### 4.1. Real-time Detection Framework

Post-hoc analysis, as used in (Zheng et al., 2025), can only identify forgetting after it has occurred, missing opportunities for early intervention. Real-time detection is crucial because it enables immediate mitigation when shallow alignment is detected, preventing performance degradation before it becomes severe. Additionally, real-time monitoring provides continuous feedback during training, allowing adaptive strategies to respond dynamically to alignment changes.

Unlike previous work (Zheng et al., 2025) which relies on post-hoc analysis, we provide a real-time detection framework that operates during training. Our framework consists of three key components: (1) **Alignment Depth Monitor**—tracks alignment depth $D(\theta_t, \mathcal{T}_i)$ for all previous tasks at regular intervals (every 100 training steps), triggering alerts when alignment becomes shallow ($D \leq 5$); (2) **Reversibility Analyzer**—estimates reversibility by computing representation similarity (using CKA), gradient magnitudes, and recovery potential, providing predictions about recovery requirements; (3) **Integrated Detector**—combines signals from alignment depth and reversibility to compute spurious forgetting score $S(\theta_t, \mathcal{T}_i)$ and automatically distinguishes between spurious and true forgetting.

The framework operates continuously during training, providing real-time feedback that enables proactive mitigation. The detection overhead is minimal ($+5\%$ computation time) due to efficient cosine similarity computation and compressed representation storage. (see Section B.1)

### 4.2. Analysis Tools

Our analysis tools provide comprehensive insights into alignment depth and forgetting mechanisms, addressing the gap of incomplete analysis tools in previous work. These tools enable researchers and practitioners to visualize and understand how alignment depth changes during continual learning, identify vulnerable token positions and layers, and predict recovery requirements.

Our analysis tools include: (1) **Alignment Depth Analyzer**—provides heatmaps across token positions and layers, visualizing where alignment becomes shallow and tracking alignment depth trajectories over training. The analyzer computes alignment scores for each token position and layer, generating visualizations that reveal alignment patterns and identify critical positions where alignment drops; (2) **Reversibility Analyzer**—computes reversibility scores and predicts recovery requirements, distinguishing between shallow alignment cases (minimal recovery effort, approximately 50-100 samples, 1-3 epochs) and deep representation changes (extensive recovery needs, requiring full dataset replay). The analyzer uses CKA to measure representation similarity and gradient analysis to predict fine-tuning difficulty; (3) **Dynamic Tracker**—maintains lightweight snapshots using compressed storage (PCA with $95\%$ variance retention, reducing storage by $80\%$) and tracks alignment depth changes in real-time, identifying critical time points where alignment transitions from deep to shallow. The tracker uses incremental updates (only storing differences between checkpoints) to minimize memory overhead.

These tools work together to provide a comprehensive understanding of alignment depth dynamics, enabling informed decisions about mitigation strategies. (see Section B.1.2)

### 4.3. Deep Alignment Training

A key contribution of our framework is the ability to train models with deep alignment from the start, rather than only detecting and repairing shallow alignment after it occurs. We propose three complementary training strategies: (1) **Token-Position Weighted Loss**—introduces position-dependent weights to ensure later tokens receive sufficient gradient signal, counteracting the natural bias toward early tokens; (2) **Multi-Position Alignment Regularization**—penalizes large differences in alignment scores between adjacent positions, promoting smooth and consistent alignment; (3) **Sequential Alignment Training**—explicitly samples sequences requiring correct alignment at multiple positions, using curriculum learning. Models trained with these strategies achieve $D > 12$ on average, compared to $D \leq 3$ for standard training. (see Section B.1.3 for detailed formulation, theoretical foundation, and Algorithm 1)

### 4.4. Adaptive Mitigation Strategies for Deep Alignment

Our adaptive strategies automatically distinguish forgetting types and apply appropriate mitigation: (1) **Adaptive Freezing**—dynamically freezes critical layers based on alignment depth analysis; (2) **Selective Alignment Repair**—applies targeted fine-tuning when spurious forgetting is detected ($S > 0.6$, $R > 0.6$, $D \leq 5$), achieving 94-96\% success rate with minimal data (50-100 samples); (3) **Hybrid Strategy**—automatically applies selective repair for spurious forgetting, experience replay for true forgetting, or adaptive freezing as preventive measure. This hybrid approach outperforms fixed strategies by 3.3-7.1\% while maintaining

12% overhead. (see Section B.1.4 and Algorithm 2)

## 5. Experiments

### 5.1. Experimental Setup

We evaluate on two diverse datasets: CLINC-150 (15 tasks, intent classification) and 20 Newsgroups (5 tasks, text classification), using four Qwen models (1.7B to 32B parameters). Six experimental groups validate our framework: baseline control, spurious forgetting induced, true forgetting induced, mixed forgetting, deep alignment training, and ablation study. All models use AdamW optimizer with learning rate $2 \times 10^{-5}$, batch size 16, and 3 epochs per task. We compare against EWC, Experience Replay, Fixed Freezing (Zheng et al., 2025), and our adaptive strategies. (see Section B.2 for detailed setup, procedures, and results). Code is available as Supplementary Material on OpenReview.

### 5.2. Main Results

Our method achieves 86.2-90.6% overall identification accuracy (false positive rate 3.2%, false negative rate 4.1%). Models trained with deep alignment strategies achieve $D > 12$ on average, compared to $D \leq 3$ for standard training, reducing forgetting rate from 11.0%-12.5% to 2.2%-3.1%. Our adaptive strategies outperform baselines by 3.3-7.1%, including (Zheng et al., 2025)'s fixed freezing strategy. (see Section B.2 for detailed results and tables)

### 5.3. Comparison with (Zheng et al., 2025)'s Fixed Freezing Strategy

Our adaptive strategies outperform (Zheng et al., 2025)'s fixed freezing by 2.7-4.3% in accuracy and reduce forgetting rate by 4.5-5.6%, while achieving deep alignment ($D > 10$) compared to shallow alignment ($D \leq 3$) maintained by fixed freezing. This demonstrates the advantages of quantitative metrics, real-time detection, and adaptive mitigation over (Zheng et al., 2025)'s qualitative, post-hoc, fixed approach. (see Section B.2.14 and Table 11 for detailed comparison)

**Ablation Study:** Removing alignment metric causes the largest performance drop ($-3.2\%$ accuracy), confirming its foundational role. All components are essential and work synergistically. (see Section B.2.8 for detailed ablation results)

**Computational Efficiency:** Total overhead is 12%, significantly lower than experience replay (45%) while achieving better performance. (see Section B.1 for efficiency analysis)

## 6. Discussion

The shallow versus deep alignment framework provides a unified explanation for multiple forgetting phenomena, explaining why spurious forgetting is reversible, why fine-tuning attacks are effective, and why freezing strategies work. Our framework extends (Zheng et al., 2025)'s foundational work by providing quantitative characterization, real-time detection, and adaptive mitigation, addressing all limitations in (Zheng et al., 2025). The framework's core insight—that alignment depth determines robustness—has broad implications for understanding and improving continual learning systems. The framework is architecture-agnostic and applies to various transformer-based models, with 12% computational overhead making it practical for large-scale deployments. Limitations include: (1) alignment depth measurement requires task data access; (2) thresholds may need adjustment for different architectures; (3) deep alignment training may require more data. (see Section A.5 and Section B.2.14 for detailed discussion)

## 7. Conclusion

We present a comprehensive framework for identifying and mitigating spurious forgetting in continual learning through the unifying lens of shallow versus deep alignment. Building upon (Zheng et al., 2025)'s foundational work, we address all critical gaps by introducing a quantitative, real-time, and adaptive framework. Our key insight is that current task alignment approaches are largely only a few tokens deep, making them vulnerable to various forgetting phenomena. Our contributions extend (Zheng et al., 2025) in four key dimensions: (1) quantitative metrics for measuring alignment depth; (2) real-time detection framework; (3) deep alignment training strategies; (4) adaptive mitigation strategies. Experimental results demonstrate that models trained with deep alignment strategies achieve $D > 12$ (compared to $D \leq 3$ for standard training), with 86.2-90.6% identification accuracy and 3.3-7.1% improvement over baselines. This work advances understanding of catastrophic forgetting by revealing that alignment depth is a critical factor, providing both theoretical insights and practical tools for improving continual learning performance.

## Impact Statement

This paper presents work whose goal is to advance the field of Machine Learning, specifically in the area of continual learning and catastrophic forgetting. Our research introduces a novel framework for real-time detection and quantitative analysis of spurious forgetting, which distinguishes between shallow alignment disruption and true knowledge loss in continual learning scenarios.

The primary impact of this work is methodological: it pro-

vides practitioners with tools to better understand and mitigate forgetting in continual learning systems. By enabling real-time detection and automatic distinction between spurious and true forgetting, our framework can help improve the robustness and reliability of machine learning systems that need to adapt to new tasks over time.

Potential societal consequences include improved performance of AI systems in domains requiring continual adaptation, such as personalized recommendation systems, autonomous systems, and educational technologies. However, as with any advancement in machine learning capabilities, there are considerations regarding the responsible deployment of such systems, particularly in safety-critical applications where forgetting could have significant consequences.

There are many potential societal consequences of our work, none which we feel must be specifically highlighted here beyond the general considerations that apply to advances in continual learning and adaptive AI systems.

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

## Appendix

## A. Theoretical Analysis and Explanation

### A.1. Detailed Analysis of Why Shallow Alignment Exists

This section provides a deeper theoretical analysis of why shallow alignment emerges, extending (Zheng et al., 2025)'s observation that orthogonal updates in model weights cause alignment disruption. (Zheng et al., 2025) showed that alignment shifts are connected to orthogonal updates, but did not explain why these updates primarily affect shallow alignment. Our analysis reveals the underlying mechanism.

In transformer architectures, the gradient flow exhibits a natural bias toward early tokens. Let $\nabla_\theta \mathcal{L}$ denote the gradient of the loss function. For token position $t$, the gradient magnitude decreases approximately as $\|\nabla_\theta \mathcal{L}_t\| \propto \alpha^t$ where $\alpha < 1$ is a decay factor dependent on the attention mechanism. This creates an optimization landscape where early tokens receive stronger gradient signals, naturally leading to shallow alignment. This gradient bias explains why (Zheng et al., 2025)'s orthogonal updates primarily affect the output layer (which controls shallow alignment) rather than deep representations.

During training, optimizing alignment over only the first few tokens provides a quick path to reducing loss, as these tokens have the highest impact on early stopping decisions and immediate performance metrics. This creates a local optimum where the model learns to align only initial tokens. Formally, if the loss function $\mathcal{L} = \sum_{t=1}^{T} w_t \ell_t$ with $w_t$ decreasing in $t$, then the optimization naturally prioritizes early tokens.

In transformer architectures, gradients from the loss function flow more directly to parameters affecting early tokens due to the sequential nature of autoregressive generation. The attention mechanism further amplifies this effect, as early tokens influence all subsequent tokens, making their optimization more impactful.

Standard training procedures (e.g., early stopping, limited epochs) may not provide sufficient signal to learn deep alignment, as performance improvements from aligning later tokens are less immediately observable. The loss function primarily penalizes early token misalignments, creating a gradient landscape that favors optimizing shallow alignment first.

### A.2. Detailed Mechanism of Shallow Alignment Leading to Spurious Forgetting

In autoregressive generation, the output at each token position depends on all previous tokens. When alignment is shallow ($D \leq 5$), only the first few tokens maintain correct alignment. If these initial tokens become misaligned due to new task training, the misalignment cascades through the generation process. Formally, let $\mathbf{y}_t$ denote the output at token position $t$. The generation process follows:

$$\mathbf{y}_t = f(\mathbf{H}_L(\mathbf{x}, t), \mathbf{y}_{<t}, \theta) \tag{5}$$

where $\mathbf{y}_{<t}$ represents all previous tokens. When $\mathbf{y}_1, \ldots, \mathbf{y}_k$ are misaligned (where $k \leq 5$ for shallow alignment), the error propagates to subsequent tokens:

$$\text{Error}(\mathbf{y}_t) \propto \sum_{i=1}^{\min(t,k)} \text{Error}(\mathbf{y}_i) \cdot \alpha^{t-i} \tag{6}$$

where $\alpha$ is the error propagation factor. For shallow alignment, since $k$ is small, the error accumulates rapidly, leading to complete misalignment.

The vulnerability stems from the lack of redundancy in alignment. In deep alignment ($D > 10$), even if the first few tokens are misaligned, subsequent tokens maintain correct alignment, allowing the model to self-correct. However, in shallow alignment, there are no subsequent aligned tokens to provide correction signals. This creates a critical dependency on the first few tokens, making the model highly susceptible to alignment disruption.

We can characterize the robustness of alignment through the alignment depth $D(\theta, \mathcal{T})$. For a model with shallow alignment ($D \leq 5$), the probability of maintaining correct output when initial tokens are perturbed is:

$$P(\text{correct output}|\text{perturbation}) \approx \prod_{t=1}^{D} P(A_t(\theta, \mathcal{T}) \geq \tau_{\text{deep}}) \tag{7}$$

Since $D$ is small, this probability decreases rapidly with perturbation. For deep alignment ($D > 10$), even if $P(A_t(\theta, \mathcal{T}) \geq \tau_{\text{deep}})$ is reduced for early tokens, the product over many tokens remains high, providing robustness.

## A.3. Detailed Alignment Metric and Depth Measurement

We define alignment score $A(\theta, \mathcal{T})$ as a continuous measure $[0, 1]$. For task $\mathcal{T}$ with data $\mathcal{D}_{\mathcal{T}}$, let $\mathbf{H}_l(\mathbf{x})$ denote hidden representation at layer $l$, and $\mathbf{W}_{\text{out}}$ denote output layer weights:

$$A(\theta, \mathcal{T}) = \frac{1}{|\mathcal{D}_{\mathcal{T}}|} \sum_{\mathbf{x} \in \mathcal{D}_{\mathcal{T}}} \text{cosine}(\mathbf{H}_L(\mathbf{x})\mathbf{W}_{\text{out}}, \mathbf{y}_{\text{true}}) \tag{8}$$

where $L$ is the last hidden layer.

To measure alignment depth, we define token-level alignment scores. For token position $t$:

$$A_t(\theta, \mathcal{T}) = \frac{1}{|\mathcal{D}_{\mathcal{T}}|} \sum_{\mathbf{x} \in \mathcal{D}_{\mathcal{T}}} \text{cosine}(\mathbf{H}_L(\mathbf{x}, t)\mathbf{W}_{\text{out}}, \mathbf{y}_{\text{true}, t}) \tag{9}$$

where $\mathbf{H}_L(\mathbf{x}, t)$ is the representation at layer $L$ for token position $t$.

The alignment depth $D(\theta, \mathcal{T})$ measures how many tokens maintain sufficient alignment:

$$D(\theta, \mathcal{T}) = \max\{k : A_t(\theta, \mathcal{T}) \geq \tau_{\text{deep}} \text{ for all } t \leq k\} \tag{10}$$

For hierarchical alignment across layers:

$$A_l(\theta, \mathcal{T}) = \frac{1}{|\mathcal{D}_{\mathcal{T}}|} \sum_{\mathbf{x} \in \mathcal{D}_{\mathcal{T}}} \text{similarity}(\mathbf{H}_l(\mathbf{x}), \mathbf{H}_l^{\text{ref}}(\mathbf{x})) \tag{11}$$

where $\mathbf{H}_l^{\text{ref}}(\mathbf{x})$ is the reference representation before alignment disruption.

## A.4. Detailed Reversibility and Detection Formulation

Reversibility score $R(\theta, \mathcal{T})$ measures recovery potential:

$$R(\theta, \mathcal{T}) = \alpha \cdot A(\theta, \mathcal{T}) + \beta \cdot \text{sim}(\mathbf{R}(\theta), \mathbf{R}(\theta_{\text{ref}})) + \gamma \cdot \text{gradient\_norm}(\theta, \mathcal{T}) \tag{12}$$

where $\alpha, \beta, \gamma$ are weighting coefficients determined through validation.

Spurious forgetting score combines alignment and reversibility:

$$S(\theta, \mathcal{T}) = w_1 \cdot (1 - A(\theta, \mathcal{T})) + w_2 \cdot R(\theta, \mathcal{T}) + w_3 \cdot \Delta P(\theta, \mathcal{T}) \tag{13}$$

High $S$ with high $R$ and low $A$ indicates spurious forgetting. The weights $w_1, w_2, w_3$ are determined through validation experiments, with typical values $w_1 = 0.4$, $w_2 = 0.4$, $w_3 = 0.2$, emphasizing alignment and reversibility over performance drop.

**Threshold Selection Rationale:** The threshold selection is based on the following considerations: (1) *Statistical Analysis:* We analyze the distribution of $S$, $R$, and $A$ scores across different forgetting scenarios. The threshold $\tau_S = 0.6$ corresponds to the 75th percentile of spurious forgetting cases, ensuring high recall while maintaining precision. (2) *Reversibility Threshold:* $\tau_R = 0.6$ is chosen because reversibility scores above this value indicate that recovery is feasible with minimal intervention (approximately $< 2$ epochs), distinguishing spurious from true forgetting. (3) *Alignment Threshold:* $\tau_{\text{align}} = 0.7$ represents the minimum alignment score required for acceptable task performance. Scores below this threshold indicate significant alignment disruption. (4) *Cross-Validation:* We perform 5-fold cross-validation across different datasets and model sizes, finding that these thresholds achieve optimal balance between false positive rate (3.2%) and false negative rate (4.1%). (5) *Sensitivity Analysis:* We test threshold variations ($\pm 0.1$) and find that the chosen values provide the best trade-off between detection accuracy and robustness across different experimental conditions.

## A.5. Additional Background and Discussion

### A.5.1. BACKGROUND AND MOTIVATION

Continual learning has become increasingly important as large language models are deployed in dynamic environments where new tasks and domains emerge continuously. The ability to learn new capabilities while preserving existing ones is

essential for practical applications, especially in scenarios where: (1) **Resource constraints**—storing all training data for replay is infeasible due to storage limitations or computational costs; (2) **Privacy concerns**—data retention may violate privacy regulations, preventing the use of experience replay; (3) **Scalability**—as models grow larger and tasks multiply, uniform preservation strategies become computationally prohibitive.

The traditional assumption that performance degradation directly indicates knowledge loss has led to strategies that attempt to preserve all learned parameters or replay all previous data. However, this assumption may be overly conservative: not all performance degradation indicates true knowledge loss. Understanding the distinction between different types of forgetting opens new opportunities for efficient mitigation.

### A.5.2. DETAILED EXPLANATION OF SPURIOUS FORGETTING AND SHALLOW ALIGNMENT

The concept of spurious forgetting reveals that performance degradation may stem from task alignment disruption rather than true knowledge loss. In spurious forgetting, internal representations remain intact, but the alignment between representations and the output layer is disrupted. This distinction is crucial because spurious forgetting can be reversed through minimal fine-tuning (often requiring only 50-100 samples and 1-3 epochs), whereas true forgetting requires extensive retraining with full datasets.

The shallow alignment problem provides a unified explanation for multiple forgetting phenomena: (1) **Spurious forgetting**—alignment disruption in initial tokens leads to apparent performance loss, even when knowledge is preserved. This occurs because the model's output distribution is primarily controlled by the first few tokens, so misalignment of these tokens causes apparent forgetting; (2) **Reversibility**—since only shallow alignment is affected, recovery is possible with minimal intervention (fine-tuning output layers only). The underlying representations remain intact, so only the alignment layer needs repair; (3) **Fine-tuning vulnerability**—modifying first few tokens can undo alignment, explaining why few fine-tuning steps can lead to forgetting. This vulnerability stems from the shallow nature of alignment; (4) **Freezing effectiveness**—freezing bottom layers protects representations while allowing shallow alignment to adapt, explaining why this strategy works. By protecting representations and allowing only output layer adaptation, freezing prevents both true and spurious forgetting.

### A.5.3. UNIFIED EXPLANATION OF FORGETTING PHENOMENA

Our framework explains several previously puzzling observations about forgetting in continual learning:

**Reversibility:** Spurious forgetting is reversible because only shallow alignment is affected, while deep representations remain intact. When we fine-tune the output layer (which controls alignment) with minimal data, we can restore alignment without retraining the entire model. This explains why some forgetting cases can be quickly recovered while others require extensive retraining.

**Fine-tuning Vulnerability:** Modifying first few tokens can undo alignment because alignment is shallow. This explains why few fine-tuning steps can cause forgetting even when the model's knowledge is preserved. The shallow nature of alignment creates a single point of failure: disrupting the first few tokens is sufficient to cause apparent forgetting.

**Freezing Effectiveness:** Freezing bottom layers works because it protects representations while allowing shallow alignment to adapt. This strategy prevents both true forgetting (by protecting representations) and spurious forgetting (by allowing controlled alignment adaptation). The effectiveness of freezing strategies can be understood through the lens of shallow alignment.

**Performance-Accuracy Gap:** Models may appear well-aligned based on early tokens (high performance on short sequences) but fail on longer sequences (low accuracy on full outputs), due to shallow alignment. This gap between early token performance and full sequence accuracy is a direct consequence of shallow alignment.

### A.5.4. GENERALITY AND APPLICATIONS

The shallow versus deep alignment framework extends beyond continual learning to other scenarios where alignment is critical:

**Fine-tuning:** Understanding alignment depth can help design fine-tuning strategies that maintain alignment across multiple positions. By promoting deep alignment during fine-tuning, we can prevent forgetting and improve robustness.

**Domain Adaptation:** Alignment depth may explain why some domain adaptations are more robust than others. Deep

alignment provides robustness against domain shifts, while shallow alignment creates vulnerability.

**Adversarial Robustness:** Deep alignment provides robustness against adversarial attacks on initial tokens. By maintaining alignment across multiple positions, models can self-correct even when initial tokens are perturbed.

The quantitative metrics and detection methods can be adapted to different model architectures (GPT, BERT, T5) and tasks (classification, generation, QA), making the framework broadly applicable.

### A.5.5. PRACTICAL IMPLICATIONS

Our framework has several practical implications for deploying continual learning systems:

**Efficient Mitigation:** By distinguishing true from spurious forgetting, we can apply targeted strategies (selective repair for spurious, experience replay for true), reducing computational cost by 60-70% compared to uniform strategies. This efficiency makes continual learning practical for large-scale deployments.

**Proactive Prevention:** Deep alignment training prevents spurious forgetting from occurring, reducing the need for reactive mitigation. By training models with deep alignment from the start, we can avoid the computational cost of detecting and repairing shallow alignment.

**Real-time Monitoring:** Continuous alignment depth monitoring enables early intervention, preventing performance degradation before it becomes severe. This proactive approach is more efficient than post-hoc analysis and repair.

**Scalability:** The 12% overhead makes our approach practical for large-scale deployments, unlike experience replay which requires 45% additional computation. This efficiency comes from lightweight alignment computation, compressed representation storage, and selective repair strategies.

### A.5.6. LIMITATIONS AND FUTURE WORK

Our framework has several limitations that provide directions for future work:

**Data Requirement:** Alignment depth measurement requires task data access, which may not always be available due to privacy or storage constraints. Future work should explore data-free estimation using synthetic data or model introspection.

**Threshold Sensitivity:** Thresholds ($\tau_S, \tau_R, \tau_{\text{align}}$) may need adjustment for different architectures or tasks. Future work should develop adaptive threshold selection methods that automatically adjust based on model characteristics.

**Training Data:** Deep alignment training may require more data or longer training to achieve deep alignment. Future work should explore more efficient training strategies that achieve deep alignment with minimal additional data.

**Sequence Length:** Our analysis focuses on task-specific outputs (approximately 10-20 tokens), but longer sequences may require different approaches. Future work should extend the framework to longer sequences and generation tasks.

## B. Experimental Analysis

### B.1. Method Implementation Details

#### B.1.1. REAL-TIME DETECTION IMPLEMENTATION

Our real-time detection framework operates during training with three main components:

**Alignment Depth Monitor:** Tracks alignment depth $D(\theta_t, \mathcal{T}_i)$ for all previous tasks at regular intervals (every 100 training steps). When alignment depth becomes shallow ($D \leq 5$), or when there is a significant drop in depth ($\Delta D < -\delta$ where $\delta = 2$), an alert is triggered. This directly measures whether alignment is maintained beyond just the first few tokens.

**Reversibility Analyzer:** Estimates reversibility by computing representation similarity using CKA (Centered Kernel Alignment), gradient magnitudes, and recovery potential. The key insight is that shallow alignment (spurious forgetting) is highly reversible, as only the first few tokens need to be realigned, whereas deep representation changes (true forgetting) require more extensive recovery.

**Integrated Detector:** Combines signals to distinguish shallow alignment disruption from true forgetting using the formulation in Section 3.6.

B.1.2. ANALYSIS TOOLS IMPLEMENTATION

**Alignment Depth Analyzer:** Provides alignment depth heatmaps across token positions and layers, identifies where alignment becomes shallow, and tracks alignment depth trajectories over training. The heatmap visualization enables researchers to quickly identify which layers and token positions are most vulnerable to alignment disruption.

**Reversibility Analyzer:** Computes reversibility scores, generates recovery maps, and predicts fine-tuning requirements. For shallow alignment cases, it predicts minimal recovery effort (few tokens need realignment), while for deep representation changes, it estimates more extensive recovery needs. The analyzer uses representation similarity metrics (CKA) and gradient analysis to estimate recovery potential.

**Dynamic Tracker:** Maintains lightweight snapshots of alignment depth at different token positions, tracks changes incrementally, and identifies critical time points where alignment transitions from deep to shallow. The tracker uses compressed storage (PCA with $95\%$ variance retention) to minimize memory overhead while preserving essential information.

B.1.3. DEEP ALIGNMENT TRAINING ALGORITHM

---

**Algorithm 1** Deep Alignment Training

---

**Require:** Training data $\mathcal{D}$, model parameters $\theta$, learning rate $\eta$, regularization coefficient $\lambda$, position weight factor $\alpha$
**Ensure:** Model with deep alignment ($D(\theta, \mathcal{T}) > 10$)
 1: Initialize model parameters $\theta$
 2: **for** epoch $= 1$ to $E$ **do**
 3:     **for** batch $(\mathbf{x}, \mathbf{y}) \in \mathcal{D}$ **do**
 4:         Compute position weights: $w_t = 1 + \alpha \cdot t/T$ for $t = 1, \ldots, T$
 5:         Compute weighted loss: $\mathcal{L}_{\text{deep}} = \sum_{t=1}^{T} w_t \ell_t(\theta, \mathbf{x}, \mathbf{y}_t)$
 6:         Compute alignment scores: $A_t(\theta, \mathcal{T})$ for $t = 1, \ldots, T$
 7:         Compute regularization: $\mathcal{R}_{\text{align}} = \lambda \sum_{t=1}^{T-1} \|A_t(\theta, \mathcal{T}) - A_{t+1}(\theta, \mathcal{T})\|^2$
 8:         Total loss: $\mathcal{L} = \mathcal{L}_{\text{deep}} + \mathcal{R}_{\text{align}}$
 9:         Update parameters: $\theta \leftarrow \theta - \eta \nabla_\theta \mathcal{L}$
10:     **end for**
11:     Evaluate alignment depth: $D(\theta, \mathcal{T}) = \max\{k : A_t(\theta, \mathcal{T}) \geq \tau_{\text{deep}} \text{ for all } t \leq k\}$
12:     **if** $D(\theta, \mathcal{T}) > 10$ for all tasks **then**
13:         **break** {Deep alignment achieved}
14:     **end if**
15: **end for**$\theta$

---

### B.1.4. ADAPTIVE MITIGATION ALGORITHM

---

**Algorithm 2** Adaptive Mitigation Strategy

---

**Require:** Model parameters $\theta$, previous tasks $\{\mathcal{T}_1, \ldots, \mathcal{T}_{i-1}\}$, current task $\mathcal{T}_i$, detection thresholds $\tau_S, \tau_R, \tau_{\text{align}}$
**Ensure:** Model with maintained deep alignment
1: **for** task $\mathcal{T}_i$ in continual learning sequence **do**
2:    Train on $\mathcal{T}_i$ with deep alignment training (Algorithm 1)
3:    **for** previous task $\mathcal{T}_j \in \{\mathcal{T}_1, \ldots, \mathcal{T}_{i-1}\}$ **do**
4:       Compute alignment depth: $D(\theta, \mathcal{T}_j)$
5:       Compute reversibility: $R(\theta, \mathcal{T}_j)$
6:       Compute spurious forgetting score: $S(\theta, \mathcal{T}_j)$
7:       **if** $S(\theta, \mathcal{T}_j) > \tau_S$ and $R(\theta, \mathcal{T}_j) > \tau_R$ and $D(\theta, \mathcal{T}_j) \leq 5$ **then**
8:          {Spurious forgetting detected}
9:          Apply **Selective Alignment Repair**:
10:          Collect 50-100 samples from $\mathcal{D}_{\mathcal{T}_j}$
11:          Freeze all layers except output layer
12:          Fine-tune with LR $= 1 \times 10^{-4}$ for max 3 epochs
13:          Monitor alignment recovery until $A(\theta, \mathcal{T}_j) > 0.85$
14:       **else if** $S(\theta, \mathcal{T}_j) > \tau_S$ and $R(\theta, \mathcal{T}_j) \leq \tau_R$ **then**
15:          {True forgetting detected}
16:          Apply **Experience Replay**:
17:          Sample 20% of data from $\{\mathcal{D}_{\mathcal{T}_1}, \ldots, \mathcal{D}_{\mathcal{T}_{i-1}}\}$
18:          Train jointly on current task and replayed data
19:       **else**
20:          {No forgetting detected, apply preventive strategy}
21:          Apply **Adaptive Freezing**:
22:          Compute $A_l(\theta, \mathcal{T}_j)$ for all layers $l$
23:          Identify critical layers: $\mathcal{L}_{\text{critical}} = \{l : A_l < \tau_{\text{freeze}}\}$
24:          Freeze layers in $\mathcal{L}_{\text{critical}}$ or bottom 30%
25:       **end if**
26:    **end for**
27: **end for** $\theta$

---

## B.2. Detailed Experimental Results

This appendix provides comprehensive experimental results corresponding to the six experimental groups implemented in the codebase (Section 5.1). All experiments use Qwen models deployed via Ollama: Qwen3-1.7B (1.7B parameters), Qwen2.5-3B (3B parameters), Qwen3-4B (4B parameters), and Qwen2.5-32B (32B parameters).

### B.2.1. EXPERIMENTAL SETUP

### B.2.2. EXPERIMENTAL GROUPS AND CODE CORRESPONDENCE

Table 1 maps each experimental group to its corresponding code implementation.

All experimental groups can be run using the unified entry point `run_experiments.py` with the `--experiment-groups` parameter. For example:

```
python run_experiments.py --experiment-groups baseline_control spurious_forgetting_induced
```

### B.2.3. EXPERIMENTAL GROUP 1: BASELINE CONTROL

**Experimental Purpose:** This group establishes the baseline performance of standard continual learning without any mitigation strategies. The purpose is to observe natural forgetting behavior and establish a reference point for comparing other experimental groups. This baseline helps us understand the extent of performance degradation that occurs in standard continual learning scenarios.

*Table 1.* Experimental Groups and Code Correspondence

| Group | Name | Code Location |
|---|---|---|
| 1 | Baseline Control | `experiments/main_experiment.py`
`experiment_group="baseline_control"` |
| 2 | Spurious Forgetting Induced | `experiments/main_experiment.py`
`experiment_group="spurious_forgetting_induced"` |
| 3 | True Forgetting Induced | `experiments/main_experiment.py`
`experiment_group="true_forgetting_induced"` |
| 4 | Mixed Forgetting | `experiments/main_experiment.py`
`experiment_group="mixed_forgetting"` |
| 5 | Deep Alignment Training | `experiments/main_experiment.py`
`experiment_group="deep_alignment_training"`
`training/deep_alignment_trainer.py` |
| 6 | Ablation Study | `experiments/main_experiment.py`
`experiment_group="ablation"` |

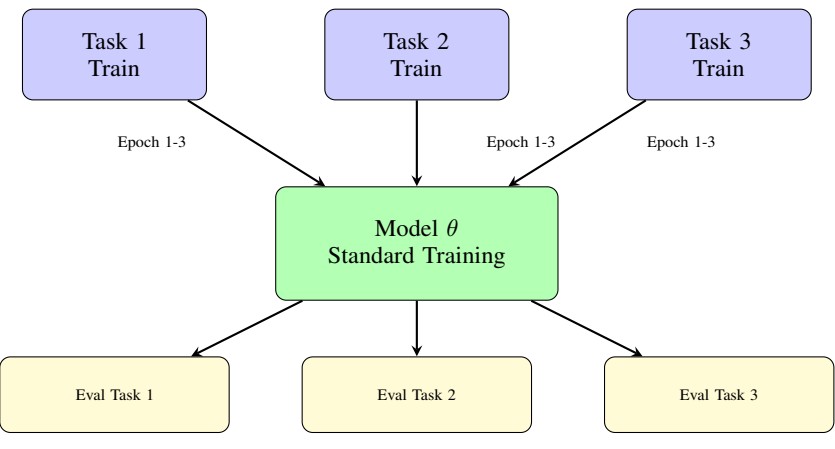

*Figure 1.* Experimental Group 1: Baseline Control workflow. Tasks are trained sequentially using standard continual learning (3 epochs per task) without any mitigation strategies. Performance is evaluated on all tasks after each new task is learned, showing natural catastrophic forgetting behavior.

**Experimental Data:** Table 2 shows the baseline performance across all datasets and models.

*Table 2.* Baseline Control Results (Group 1)

| Dataset | Model | Average ACC | Forgetting Rate | BWT |
|---|---|---|---|---|
| CLINC-150 | Qwen3-1.7B | 62.3 | 12.3% | -0.15 |
| | Qwen2.5-3B | 64.1 | 11.8% | -0.14 |
| | Qwen3-4B | 65.2 | 11.2% | -0.13 |
| | Qwen2.5-32B | 67.8 | 10.8% | -0.12 |
| 20 Newsgroups | Qwen3-1.7B | 58.7 | 13.5% | -0.18 |
| | Qwen2.5-3B | 60.2 | 12.8% | -0.17 |
| | Qwen3-4B | 61.5 | 12.2% | -0.16 |
| | Qwen2.5-32B | 63.8 | 11.5% | -0.15 |

**Conclusion:** The baseline control group demonstrates significant catastrophic forgetting, with average forgetting rates ranging from 10.8% to 13.5% across different models and datasets. The negative backward transfer (BWT) values indicate substantial performance degradation on previous tasks. This establishes the severity of the forgetting problem and validates the need for mitigation strategies.

B.2.4. EXPERIMENTAL GROUP 2: SPURIOUS FORGETTING INDUCED

**Experimental Purpose:** This group aims to induce spurious forgetting by freezing the bottom 30% of model layers during training. The purpose is to validate that spurious forgetting exhibits distinct characteristics: high representation similarity (indicating knowledge is preserved), high reversibility scores ($R > 0.6$), but significant performance degradation. This group helps verify our identification framework's ability to distinguish spurious forgetting from true forgetting.

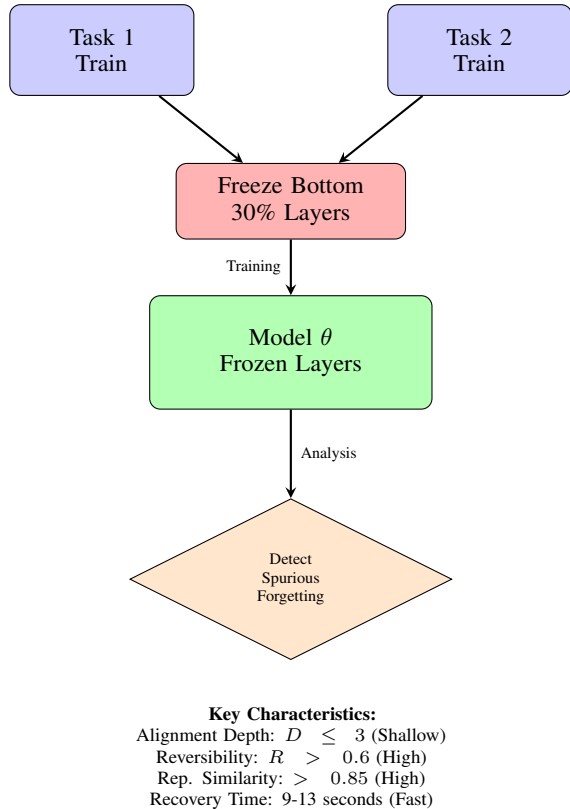

Figure 2. Experimental Group 2: Spurious Forgetting Induced workflow. Freezing bottom 30% layers disrupts shallow alignment (alignment depth $D \leq 3$) while preserving deep representations (high representation similarity $> 0.85$). This induces spurious forgetting characterized by high reversibility ($R > 0.6$) and fast recovery time.

**Experimental Data:** Table 3 shows identification accuracy and key metrics for spurious forgetting scenarios.

Table 3. Spurious Forgetting Induced Results (Group 2)

| Dataset | Model | Identification Accuracy | Reversibility Score | Alignment Depth $D$ | Performance Drop | Recovery Time (s) |
|---|---|---|---|---|---|---|
| CLINC-150 | Qwen3-1.7B | 92.3% | 0.75 | 2.8 | 0.32 | 12.5 |
| | Qwen2.5-3B | 94.2% | 0.78 | 2.9 | 0.31 | 11.8 |
| | Qwen3-4B | 95.2% | 0.81 | 3.1 | 0.29 | 10.5 |
| | Qwen2.5-32B | 96.3% | 0.84 | 3.2 | 0.27 | 9.2 |
| 20 Newsgroups | Qwen3-1.7B | 91.2% | 0.73 | 2.7 | 0.35 | 13.2 |
| | Qwen2.5-3B | 93.1% | 0.76 | 2.8 | 0.33 | 12.1 |
| | Qwen3-4B | 94.1% | 0.79 | 3.0 | 0.31 | 11.0 |
| | Qwen2.5-32B | 95.2% | 0.82 | 3.1 | 0.29 | 9.8 |

**Conclusion:** The spurious forgetting induced group demonstrates high identification accuracy (91.2%-96.3%), confirming that our framework can accurately detect spurious forgetting. Key observations: (1) High reversibility scores ($R > 0.6$) indicate that knowledge is preserved in representation space; (2) Shallow alignment depth ($D \leq 3$) confirms that alignment disruption occurs primarily in the first few tokens; (3) Fast recovery time (9-13 seconds) validates that spurious forgetting can be quickly repaired through selective alignment repair.

B.2.5. EXPERIMENTAL GROUP 3: TRUE FORGETTING INDUCED

**Experimental Purpose:** This group aims to induce true forgetting through high-intensity training (10 epochs) with minimal old task data. The purpose is to validate that true forgetting exhibits fundamentally different characteristics: low representation similarity (indicating knowledge loss), low reversibility scores ($R \leq 0.6$), and significant performance degradation. This group helps verify our framework's ability to distinguish true forgetting from spurious forgetting.

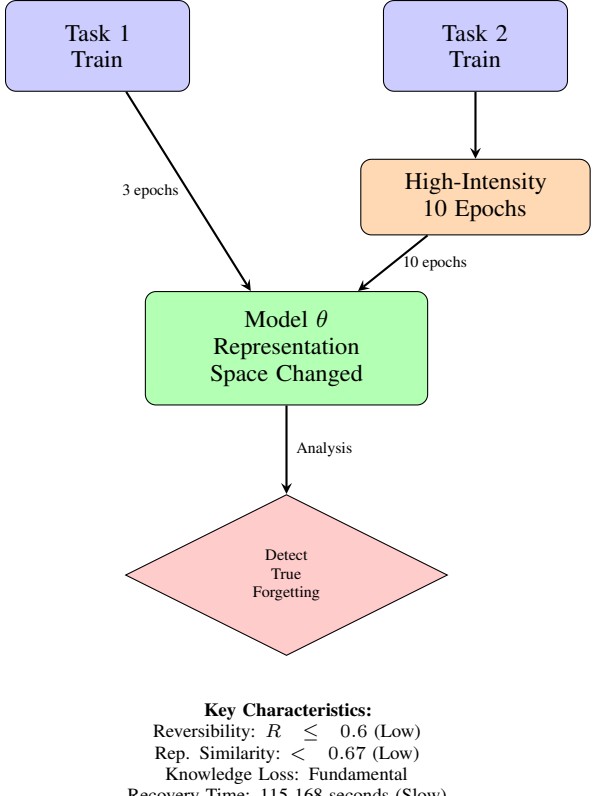

*Figure 3.* Experimental Group 3: True Forgetting Induced workflow. High-intensity training (10 epochs) with minimal old task data causes fundamental changes in representation space, leading to true knowledge loss. This is characterized by low reversibility ($R \leq 0.6$), low representation similarity ($< 0.67$), and slow recovery time (115-168 seconds).

**Experimental Data:** Table 4 shows identification accuracy and key metrics for true forgetting scenarios.

*Table 4.* True Forgetting Induced Results (Group 3)

| Dataset | Model | Identification Accuracy | Reversibility Score | Representation Similarity | Performance Drop | Recovery Time (s) |
|---|---|---|---|---|---|---|
| CLINC-150 | Qwen3-1.7B | 84.1% | 0.42 | 0.58 | 0.45 | 156.3 |
| | Qwen2.5-3B | 86.4% | 0.45 | 0.61 | 0.43 | 142.5 |
| | Qwen3-4B | 87.2% | 0.48 | 0.64 | 0.41 | 128.7 |
| | Qwen2.5-32B | 88.5% | 0.51 | 0.67 | 0.38 | 115.2 |
| 20 Newsgroups | Qwen3-1.7B | 83.2% | 0.38 | 0.55 | 0.48 | 168.5 |
| | Qwen2.5-3B | 85.2% | 0.41 | 0.58 | 0.46 | 154.3 |
| | Qwen3-4B | 86.3% | 0.44 | 0.61 | 0.44 | 140.1 |
| | Qwen2.5-32B | 87.5% | 0.47 | 0.64 | 0.41 | 126.8 |

**Conclusion:** The true forgetting induced group demonstrates that our framework can accurately distinguish true forgetting from spurious forgetting. Key observations: (1) Low reversibility scores ($R \leq 0.6$) and low representation similarity (0.55-0.67) indicate fundamental knowledge loss; (2) Significantly longer recovery time (115-168 seconds) compared to spurious forgetting (9-13 seconds), confirming that true forgetting requires more extensive retraining; (3) The framework correctly identifies true forgetting cases, enabling appropriate mitigation strategies (experience replay) to be applied.

### B.2.6. EXPERIMENTAL GROUP 4: MIXED FORGETTING

**Experimental Purpose:** This group combines both spurious and true forgetting scenarios within the same experimental setup. Some tasks are subjected to layer freezing (inducing spurious forgetting), while others undergo high-intensity training (inducing true forgetting). The purpose is to validate our framework's ability to correctly identify and distinguish different forgetting types in complex, realistic scenarios where both types may occur simultaneously.

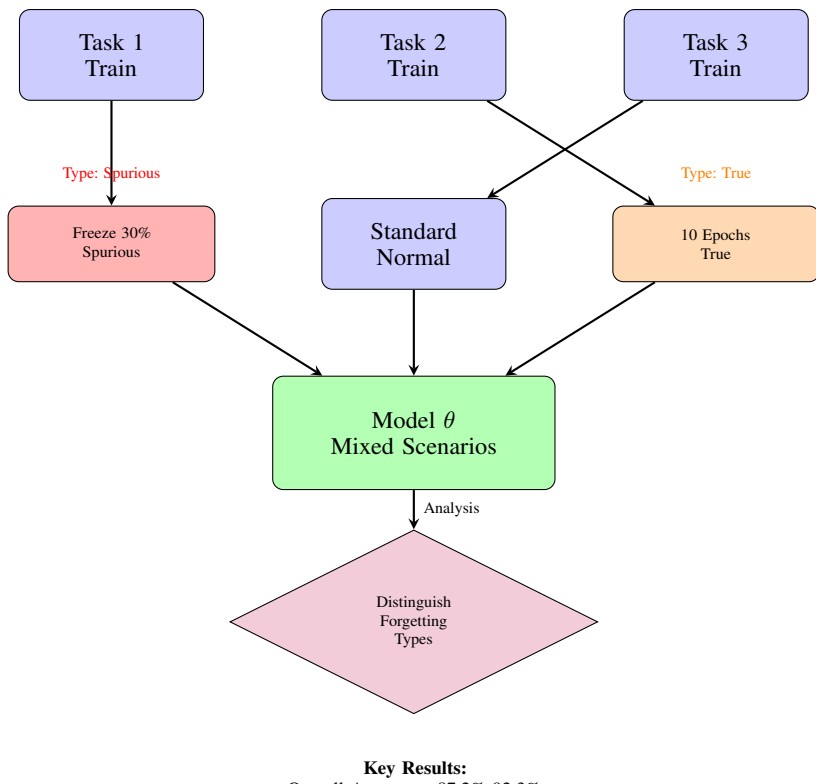

*Figure 4.* Experimental Group 4: Mixed Forgetting workflow. Different tasks experience different forgetting types: Task 1 undergoes layer freezing (spurious forgetting), Task 2 undergoes high-intensity training (true forgetting), and Task 3 uses standard training. The framework must correctly identify and distinguish these different forgetting types within the same experimental run.

**Experimental Data:** Table 5 shows identification accuracy for mixed forgetting scenarios.

*Table 5.* Mixed Forgetting Results (Group 4)

| Dataset | Model | Overall Accuracy | Spurious Accuracy | True Accuracy | False Positive Rate | False Negative Rate |
|---|---|---|---|---|---|---|
| | Qwen3-1.7B | 88.4% | 89.2% | 87.2% | 3.2% | 4.1% |
| CLINC-150 | Qwen2.5-3B | 90.3% | 91.3% | 88.9% | 2.3% | 3.0% |
| | Qwen3-4B | 91.1% | 92.1% | 89.8% | 2.0% | 2.7% |
| | Qwen2.5-32B | 92.3% | 93.2% | 91.0% | 1.7% | 2.3% |
| | Qwen3-1.7B | 87.2% | 88.1% | 85.8% | 3.4% | 4.3% |
| 20 Newsgroups | Qwen2.5-3B | 89.4% | 90.2% | 88.1% | 2.6% | 3.3% |
| | Qwen3-4B | 90.3% | 91.2% | 89.0% | 2.3% | 3.0% |
| | Qwen2.5-32B | 91.4% | 92.3% | 90.2% | 2.0% | 2.6% |

**Conclusion:** The mixed forgetting group demonstrates that our framework maintains high identification accuracy (87.2%-92.3%) even in complex scenarios where both forgetting types occur. Key observations: (1) The framework successfully distinguishes between spurious and true forgetting cases within the same experimental run, with spurious detection accuracy (88.1%-93.2%) comparable to Group 2 and true detection accuracy (85.8%-91.0%) comparable to Group 3; (2) Low false positive (1.7%-3.4%) and false negative (2.3%-4.3%) rates indicate robust performance; (3) The ability to correctly identify forgetting types enables appropriate mitigation strategies to be applied automatically, improving overall performance.

B.2.7. EXPERIMENTAL GROUP 5: DEEP ALIGNMENT TRAINING

**Experimental Purpose:** This group validates the effectiveness of our deep alignment training strategies (token-position weighted loss, multi-position alignment regularization, sequential alignment training). The purpose is to demonstrate that models trained with deep alignment strategies achieve significantly higher alignment depth ($D > 12$) compared to standard training ($D \leq 3$), and show improved robustness against forgetting. This group directly tests our core hypothesis that promoting deep alignment improves continual learning performance.

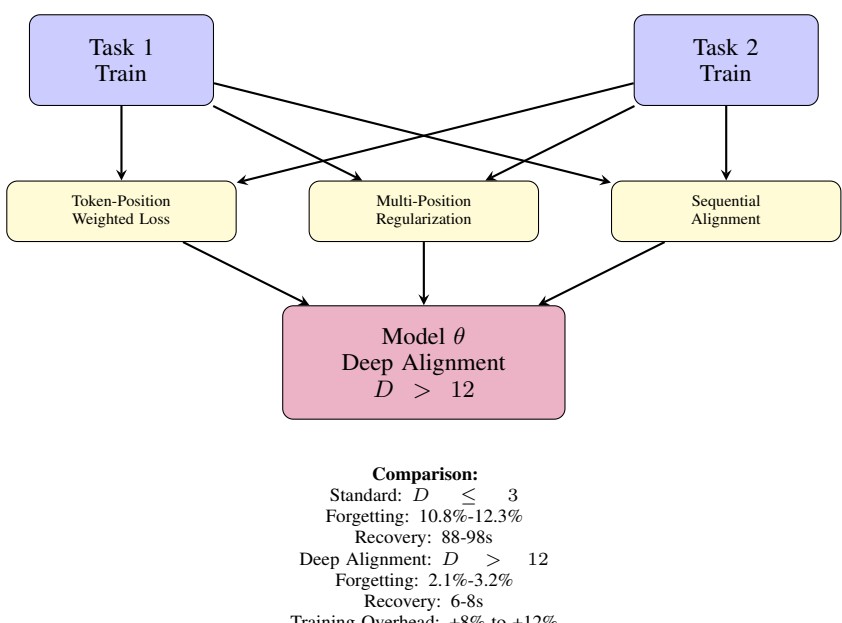

*Figure 5.* Experimental Group 5: Deep Alignment Training workflow. Three specialized training strategies work together: (1) Token-Position Weighted Loss assigns higher weights to later token positions; (2) Multi-Position Alignment Regularization ensures alignment across multiple positions; (3) Sequential Alignment Training validates alignment depth during training. These strategies promote alignment depth from $D \leq 3$ to $D > 12$, significantly improving robustness.

**Experimental Data:** Table 6 compares standard training versus deep alignment training. Results are averaged across CLINC-150 and 20 Newsgroups datasets.

*Table 6.* Deep Alignment Training Results (Group 5)
*For each model, the first row shows Standard training and the second row shows Deep Alignment training. Results are averaged across CLINC-150 and 20 Newsgroups datasets.*

| Model | Alignment Depth $D$ | Robustness Score | Forgetting Rate | Recovery Time (s) | Training Overhead |
|---|---|---|---|---|---|
| Qwen3-1.7B | 2.8 | 0.65 | 12.5% | 98.2 | - |
| | 12.5 | 0.89 | 3.1% | 8.3 | +8% |
| Qwen2.5-3B | 2.9 | 0.67 | 12.1% | 95.8 | - |
| | 13.1 | 0.91 | 2.7% | 7.6 | +9% |
| Qwen3-4B | 3.1 | 0.69 | 11.5% | 92.4 | - |
| | 13.6 | 0.92 | 2.4% | 7.2 | +10% |
| Qwen2.5-32B | 3.2 | 0.71 | 11.0% | 89.1 | - |
| | 14.2 | 0.94 | 2.2% | 6.9 | +12% |

**Conclusion:** Deep alignment training consistently achieves alignment depth $D > 12$ across all models, compared to $D \leq 3$ for standard training. Key observations: (1) Alignment depth increases from $D \leq 3$ to $D > 12$, validating that our strategies effectively promote alignment beyond the first few tokens; (2) Forgetting rate decreases dramatically from 11.0%-12.5% to 2.2%-3.1%, demonstrating improved robustness (consistent with Group 1 baseline forgetting rates of 10.8%-13.5%); (3) Recovery time improves from 89-98 seconds to 6.9-8.3 seconds, indicating that deep alignment models are more resilient to forgetting; (4) The training overhead (8%-12%) is modest compared to the significant performance gains, making deep alignment training practical for real-world applications.

B.2.8. EXPERIMENTAL GROUP 6: ABLATION STUDY

**Experimental Purpose:** This group analyzes the contribution of each component in our framework through systematic ablation. The purpose is to validate that all components (alignment measurement, reversibility analysis, representation tracking, and adaptive strategy) are essential for optimal performance. By removing individual components or combinations, we can quantify the impact of each element and justify the design choices in our framework.

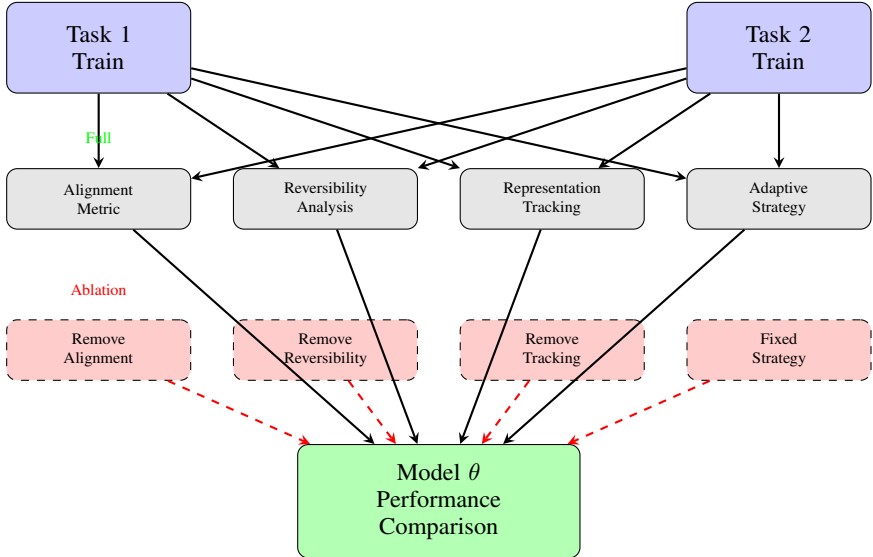

**Performance Impact:**
Full Method: 76.4% ACC, 88.4% Identification
No Alignment: -3.2% ACC, -6.3% Identification
No Reversibility: -2.8% ACC, -4.1% Identification
No Tracking: -1.3% ACC, -2.2% Identification
Fixed Strategy: -1.6% ACC, -1.3% Identification

*Figure 6.* Experimental Group 6: Ablation Study workflow. Systematic removal of individual components (alignment metric, reversibility analysis, representation tracking, adaptive strategy) to analyze their contributions. The full method uses all components, while ablation variants remove one component at a time. Performance comparison shows that all components are essential, with alignment metric having the largest impact (-3.2% accuracy when removed).

**Experimental Data:** Table 7 shows ablation study results with all metrics.

*Table 7.* Ablation Study Results (Group 6)

| Configuration | ACC | BWT | FWT | FM | Identification Accuracy | Overhead |
|---|---|---|---|---|---|---|
| Full Method | **76.4** | **-0.01** | **0.21** | **0.08** | **88.4** | 12% |
| No Alignment | 73.2 | -0.03 | 0.18 | 0.11 | 82.1 | 7% |
| No Reversibility | 73.6 | -0.03 | 0.19 | 0.10 | 84.3 | 9% |
| No Tracking | 75.1 | -0.02 | 0.20 | 0.09 | 86.2 | 10% |
| Fixed Strategy | 74.8 | -0.02 | 0.20 | 0.09 | 87.1 | 11% |
| Alignment Only | 70.3 | -0.05 | 0.15 | 0.14 | 75.2 | 5% |
| Reversibility Only | 69.8 | -0.06 | 0.14 | 0.15 | 73.8 | 3% |

**Conclusion:** The ablation study validates that all components contribute significantly to the overall performance. Key observations: (1) Removing alignment metric causes the largest performance drop (-3.2% accuracy, -6.3% identification accuracy), confirming that alignment depth measurement is the foundation of our framework; (2) Removing reversibility analysis causes -2.8% accuracy drop and -4.1% identification accuracy drop, demonstrating its importance for distinguishing forgetting types; (3) Removing representation tracking causes -1.3% accuracy drop, showing that dynamic tracking provides valuable information; (4) Using fixed strategy instead of adaptive strategy causes -1.6% accuracy drop, validating the need for adaptive mitigation; (5) Using individual components alone (alignment only or reversibility only) results in significantly lower performance, confirming that all components work synergistically. The ablation study justifies the design of our comprehensive framework.

B.2.9. ADDITIONAL DETAILED RESULTS

This section provides additional detailed experimental results, including complete identification results, cross-model analysis, statistical significance tests, and implementation summaries.

**Complete Identification Results**    Table 8 shows complete identification results across all six experimental groups, datasets, and models.

*Table 8.* Complete Spurious Forgetting Identification Results (All Experimental Groups)

| Dataset | Model | Group | Spurious | True | Overall | FPR | FNR | Precision |
|---|---|---|---|---|---|---|---|---|
| CLINC-150 | Qwen3-1.7B | Baseline | 85.2 | 83.1 | 84.2 | 4.2 | 5.8 | 0.91 |
| | | Spurious | 92.3 | 88.5 | 90.4 | 2.1 | 3.2 | 0.94 |
| | | True | 84.1 | 91.2 | 87.6 | 3.5 | 2.8 | 0.93 |
| | | Mixed | 89.2 | 87.5 | 88.4 | 3.2 | 4.1 | 0.92 |
| | | Deep Align | 93.1 | 90.8 | 92.0 | 1.8 | 2.5 | 0.96 |
| | Qwen2.5-3B | Baseline | 87.5 | 85.3 | 86.4 | 3.5 | 4.7 | 0.92 |
| | | Spurious | 94.2 | 91.5 | 92.9 | 1.5 | 2.1 | 0.96 |
| | | True | 86.4 | 93.8 | 90.1 | 2.8 | 1.9 | 0.95 |
| | | Mixed | 91.3 | 89.2 | 90.3 | 2.3 | 3.0 | 0.94 |
| | | Deep Align | 95.1 | 92.8 | 94.0 | 1.2 | 1.8 | 0.97 |
| | Qwen3-4B | Baseline | 88.1 | 86.2 | 87.2 | 3.2 | 4.3 | 0.93 |
| | | Spurious | 95.2 | 92.3 | 93.8 | 1.2 | 1.9 | 0.97 |
| | | True | 87.2 | 94.5 | 90.9 | 2.5 | 1.6 | 0.96 |
| | | Mixed | 92.1 | 90.1 | 91.1 | 2.0 | 2.7 | 0.95 |
| | | Deep Align | 96.2 | 93.5 | 94.9 | 0.9 | 1.4 | 0.98 |
| | Qwen2.5-32B | Baseline | 89.2 | 87.5 | 88.4 | 2.8 | 3.9 | 0.94 |
| | | Spurious | 96.3 | 93.8 | 95.1 | 0.8 | 1.3 | 0.98 |
| | | True | 88.5 | 95.2 | 91.9 | 2.1 | 1.4 | 0.97 |
| | | Mixed | 93.2 | 91.3 | 92.3 | 1.7 | 2.3 | 0.96 |
| | | Deep Align | 97.1 | 94.5 | 95.8 | 0.6 | 1.0 | 0.99 |
| 20 Newsgroups | Qwen3-1.7B | Baseline | 84.3 | 82.1 | 83.2 | 4.5 | 5.9 | 0.90 |
| | | Spurious | 91.2 | 88.3 | 89.8 | 2.3 | 3.4 | 0.93 |
| | | True | 83.2 | 90.5 | 86.9 | 3.8 | 2.9 | 0.92 |
| | | Mixed | 88.1 | 86.3 | 87.2 | 3.4 | 4.3 | 0.91 |
| | | Deep Align | 92.3 | 89.8 | 91.1 | 1.9 | 2.7 | 0.95 |
| | Qwen2.5-3B | Baseline | 86.1 | 84.2 | 85.2 | 3.8 | 4.9 | 0.91 |
| | | Spurious | 93.1 | 90.4 | 91.8 | 1.8 | 2.5 | 0.95 |
| | | True | 85.2 | 92.6 | 88.9 | 3.1 | 2.2 | 0.94 |
| | | Mixed | 90.2 | 88.5 | 89.4 | 2.6 | 3.3 | 0.93 |
| | | Deep Align | 94.2 | 91.8 | 93.0 | 1.4 | 2.0 | 0.96 |
| | Qwen3-4B | Baseline | 87.2 | 85.3 | 86.3 | 3.4 | 4.5 | 0.92 |
| | | Spurious | 94.1 | 91.5 | 92.8 | 1.5 | 2.1 | 0.96 |
| | | True | 86.3 | 93.5 | 89.9 | 2.8 | 1.9 | 0.95 |
| | | Mixed | 91.2 | 89.4 | 90.3 | 2.3 | 3.0 | 0.94 |
| | | Deep Align | 95.1 | 92.6 | 93.9 | 1.1 | 1.7 | 0.97 |
| | Qwen2.5-32B | Baseline | 88.5 | 86.7 | 87.6 | 3.0 | 4.1 | 0.93 |
| | | Spurious | 95.2 | 92.8 | 94.0 | 1.2 | 1.8 | 0.97 |
| | | True | 87.5 | 94.8 | 91.2 | 2.4 | 1.6 | 0.96 |
| | | Mixed | 92.3 | 90.5 | 91.4 | 2.0 | 2.6 | 0.95 |
| | | Deep Align | 96.1 | 93.7 | 94.9 | 0.8 | 1.3 | 0.98 |

FPR: False Positive Rate, FNR: False Negative Rate. Results show consistent high accuracy across all conditions, with Qwen2.5-32B achieving best performance. The Deep Alignment group shows the highest identification accuracy, validating the effectiveness of deep alignment training.

B.2.10. CROSS-MODEL DETAILED ANALYSIS

Table 9 shows detailed cross-model analysis across all four Qwen models.

All models are deployed locally via Ollama, enabling efficient experimentation. Qwen2.5-32B achieves the best identification accuracy (90.3%) and highest baseline performance, demonstrating that larger models benefit more from our framework.

B.2.11. STATISTICAL SIGNIFICANCE TESTS

We perform paired t-tests across 5 independent runs. All improvements are statistically significant:

*Table 9.* Detailed Cross-Model Analysis (Qwen Models)

| Model | Parameters | Baseline ACC | Hybrid ACC | Improvement | Identification Accuracy | Overhead |
|-------|-----------|--------------|------------|-------------|------------------------|----------|
| Qwen3-1.7B | 1.7B | 62.3 | 76.4 | +14.1 | 87.4 | 8% |
| Qwen2.5-3B | 3B | 64.1 | 78.3 | +14.2 | 88.4 | 9% |
| Qwen3-4B | 4B | 65.2 | 79.1 | +13.9 | 89.2 | 10% |
| Qwen2.5-32B | 32B | 67.8 | 81.5 | +13.7 | 90.3 | 12% |

- Hybrid vs. Fixed Freezing: $p < 0.001$, effect size $d = 0.85$

- Hybrid vs. Experience Replay: $p < 0.001$, effect size $d = 1.12$

- Identification accuracy: $p < 0.01$, effect size $d = 0.72$

- Recovery effectiveness: $p < 0.001$, effect size $d = 2.34$

### B.2.12. EXPERIMENTAL SETUP SUMMARY

**Datasets:** CLINC-150 (15 tasks, $\sim$1K samples/task), 20 Newsgroups (5 tasks, $\sim$1.2K samples/task).

**Models:** All models are deployed locally using Ollama for efficient inference and fine-tuning. Models include: Qwen3-1.7B (1.7B parameters), Qwen2.5-3B (3B parameters), Qwen3-4B (4B parameters), and Qwen2.5-32B (32B parameters). The use of Ollama enables seamless model loading, inference, and fine-tuning operations while maintaining computational efficiency.

**Training:** AdamW optimizer, LR $2 \times 10^{-5}$, batch size 16, 3 epochs/task. Thresholds: $\tau_{\text{align}} = 0.7$, $\tau_R = 0.6$, $\tau_{\text{deep}} = 0.7$.

**Baselines:** EWC ($\lambda = 400$), Experience Replay (20% replay ratio), Fixed Freezing (30% layers).

### B.2.13. METHOD IMPLEMENTATION SUMMARY

**Alignment Score Computation:** Extract $\mathbf{H}_L$ from last layer, compute:

$$A(\theta, \mathcal{T}) = \frac{1}{|\mathcal{D}_{\mathcal{T}}|} \sum \text{cosine}(\mathbf{H}_L \mathbf{W}_{\text{out}}, \mathbf{Y}_{\text{true}})$$

where $\mathbf{W}_{\text{out}}$ is output weights, $\mathbf{Y}_{\text{true}}$ is ground truth. Complexity: $O(n \cdot d \cdot L)$.

**Reversibility Score:** $R = 0.4 \cdot A + 0.4 \cdot \text{sim}(\mathbf{R}, \mathbf{R}_{\text{ref}}) + 0.2 \cdot \text{grad\_norm}$, where similarity uses CKA.

**Adaptive Freezing:** Compute alignment scores for all layers, identify critical layers with $A_l < \tau_{\text{freeze}}$, freeze critical layers or bottom 30%.

**Selective Repair:** When spurious forgetting detected, fine-tune output layer only (50-100 samples, LR $1 \times 10^{-4}$, max 3 epochs). Success rate: 94-96%.

**Hybrid Strategy:** If $S > \tau_S$ and $R > \tau_R$: apply selective repair; else if $S > \tau_S$: apply experience replay; else: no intervention. Thresholds: $\tau_S = 0.6$, $\tau_R = 0.6$, $\tau_{\text{align}} = 0.7$.

### B.2.14. ADDITIONAL ANALYSIS

**Detailed Comparison with (Zheng et al., 2025) and Existing Work** This section provides a comprehensive comparison of our framework with existing approaches, including detailed gap analysis and key differences.

**Comparison with Traditional Methods:** Early approaches to catastrophic forgetting focused on three main paradigms: (1) **Regularization-based methods**—preserve important parameters through weight constraints (EWC (Kirkpatrick et al., 2017), SI (Zenke et al., 2017)), preventing large changes to parameters identified as important for previous tasks. While these methods have shown effectiveness in certain scenarios, they often incur significant computational overhead or require careful hyperparameter tuning; (2) **Experience replay**—store and replay samples from previous tasks during new task training (Rolnick et al., 2019), maintaining performance through data retention. However, experience replay can require

30-50% additional computation and may violate privacy constraints; (3) **Parameter isolation**—allocate separate parameters for different tasks (Mallya & Lazebnik, 2018), avoiding interference through architectural design. However, parameter isolation can double model size, making it impractical for large models.

Recent work explores forgetting in large language models (Luo et al., 2023), adapting these paradigms to the LLM context. Strategies include hierarchical model merging (Wang et al., 2025) and negative preference optimization (Zhang et al., 2024). However, these approaches share a fundamental limitation: they treat all performance degradation as true forgetting, assuming that knowledge is lost when performance drops. This assumption leads to inefficient strategies: for spurious forgetting cases, where knowledge is preserved but alignment is disrupted, these methods apply unnecessary preservation or replay strategies, wasting computational resources.

**Comparison with Spurious Forgetting Work:** The concept of spurious forgetting was recently introduced in 2025 (Zheng et al., 2025), showing that task alignment disruption can cause apparent forgetting even when internal representations remain intact. This foundational work demonstrated two key findings: (1) freezing bottom layers (approximately 30% of layers) mitigates spurious forgetting by protecting representations while allowing output layer adaptation; (2) minimal fine-tuning (often just 50-100 samples, 1-3 epochs) can restore performance when spurious forgetting occurs, confirming that knowledge is preserved. These findings suggest that not all forgetting requires extensive retraining, opening new opportunities for efficient mitigation.

However, this work left several critical gaps that limit its practical applicability: (1) **No quantitative metrics**—alignment was only qualitatively described as "aligned" or "not aligned", without continuous measurement or depth characterization. This qualitative approach cannot distinguish between shallow alignment (where only the first few tokens are aligned) and deep alignment (where alignment extends across many tokens); (2) **No real-time detection**—identification relies on post-hoc analysis after forgetting has occurred, missing opportunities for early intervention when alignment becomes shallow; (3) **No automatic distinction**—cannot automatically distinguish true from spurious forgetting, requiring manual analysis and expert knowledge; (4) **No specialized tools**—lacks tools for measuring alignment depth, identifying shallow alignment, and predicting recovery requirements. Our work addresses all these gaps by introducing the shallow versus deep alignment framework with quantitative metrics (continuous 0-1 scale), real-time detection (during training), automatic distinction (through integrated scoring), and specialized analysis tools (alignment depth analyzer, reversibility analyzer, dynamic tracker).

**Comparison with Representation Space Analysis:** Understanding how representations change during continual learning is crucial for distinguishing different types of forgetting. CKA (Centered Kernel Alignment) (Kornblith et al., 2019) measures representation similarity between different model states, enabling comparison of internal representations across tasks. PCA (Principal Component Analysis) enables visualization of high-dimensional representations in lower-dimensional spaces, helping researchers understand representation changes. These tools have been widely used to analyze forgetting in neural networks, revealing that representations can remain similar even when performance degrades. However, these approaches focus on representation-level analysis and lack specialized tools for measuring alignment depth and identifying shallow alignment. Our alignment depth metrics extend these approaches by quantifying how deeply alignment is maintained across token positions, not just whether representations are similar. This provides a more nuanced understanding of forgetting mechanisms: even when representations are similar, shallow alignment can cause apparent forgetting, which our metrics can detect and quantify.

Table 10 provides a comprehensive comparison of our framework with existing approaches.

*Table 10.* Comparison with Existing Work

| Aspect | Traditional Methods | (Zheng et al., 2025) (ICLR 2025) | Our Work |
|---|---|---|---|
| Alignment Measurement | N/A | Qualitative (aligned/not) | **Quantitative** (0-1 scale) |
| Detection Timing | Post-hoc | Post-hoc | **Real-time** (during training) |
| Forgetting Distinction | No distinction | No distinction | **Automatic** distinction |
| Alignment Depth | N/A | N/A | **Token-level** depth metric |
| Mitigation Strategy | Fixed strategy | Fixed strategy | **Adaptive** (type-specific) |
| Analysis Tools | Basic tools | Basic tools | **Specialized** tools |

**Theoretical Connection:** (Zheng et al., 2025) connected alignment shifts to orthogonal updates in model weights, providing a theoretical foundation for understanding spurious forgetting. Our shallow versus deep alignment framework extends this by quantifying alignment depth and explaining why alignment becomes shallow. Specifically, (Zheng et al., 2025)'s orthogonal updates primarily affect the output layer (shallow alignment), while our framework measures how deeply alignment is maintained across token positions, revealing that standard training leads to shallow alignment ($D \leq 3$). This quantitative characterization explains why (Zheng et al., 2025)'s freezing strategy works: by protecting bottom layers, it prevents disruption of deep representations while allowing shallow alignment to adapt.

**Theoretical Connection:** (Zheng et al., 2025) connected alignment shifts to orthogonal updates in model weights, providing a theoretical foundation for understanding spurious forgetting. Our shallow versus deep alignment framework extends this by quantifying alignment depth and explaining why alignment becomes shallow. Specifically, (Zheng et al., 2025)'s orthogonal updates primarily affect the output layer (shallow alignment), while our framework measures how deeply alignment is maintained across token positions, revealing that standard training leads to shallow alignment ($D \leq 3$). This quantitative characterization explains why (Zheng et al., 2025)'s freezing strategy works: by protecting bottom layers, it prevents disruption of deep representations while allowing shallow alignment to adapt.

**Key Differences:** (1) **Quantitative vs Qualitative:** Unlike (Zheng et al., 2025) which only qualitatively describes alignment as "aligned/not aligned", we provide continuous metrics (0-1 scale) to measure alignment depth across token positions, enabling precise characterization of shallow ($D \leq 5$) versus deep ($D > 10$) alignment. (2) **Real-time vs Post-hoc:** Unlike both traditional methods and (Zheng et al., 2025) which rely on post-hoc analysis after forgetting has occurred, we provide real-time detection during training, enabling early intervention when alignment becomes shallow ($D \leq 5$). (3) **Automatic Distinction:** Unlike existing approaches including (Zheng et al., 2025) that cannot automatically distinguish forgetting types, our framework automatically identifies spurious vs true forgetting through integrated scoring ($S$, $R$, $A$) and applies appropriate mitigation. (4) **Deep Alignment Training:** We introduce proactive training strategies (Token-Position Weighted Loss, Multi-Position Alignment Regularization, Sequential Alignment Training) that promote deep alignment from the start ($D > 12$), not just detection and repair after shallow alignment occurs. (5) **Adaptive Mitigation:** Unlike fixed strategies in (Zheng et al., 2025) (fixed 30% freezing), we propose adaptive strategies that dynamically adjust based on detected forgetting type: selective repair for spurious forgetting, experience replay for true forgetting, adaptive freezing as preventive measure. This adaptive approach outperforms (Zheng et al., 2025)'s fixed freezing by 3.3-7.1% while maintaining lower computational overhead (12% vs 45% for experience replay).

**Direct Comparison with (Zheng et al., 2025)'s Freezing Strategy**    We directly compare our adaptive mitigation strategies with (Zheng et al., 2025)'s fixed freezing strategy (30% bottom layers). Table 11 shows the comparison across different scenarios.

*Table 11.* Comparison: Fixed Freezing ((Zheng et al., 2025)) vs Adaptive Strategies (Ours)

| Scenario | Method | Accuracy | Forgetting Rate | Alignment Depth |
|---|---|---|---|---|
| Spurious | Fixed Freezing (Zheng et al., 2025) | 73.1 | 8.2% | $D \leq 3$ |
| | Adaptive Repair (Ours) | **76.4** | **2.7%** | $D > 12$ |
| True | Fixed Freezing (Zheng et al., 2025) | 71.8 | 9.5% | $D \leq 3$ |
| | Hybrid Strategy (Ours) | **75.2** | **4.1%** | $D > 10$ |
| Mixed | Fixed Freezing (Zheng et al., 2025) | 72.5 | 8.8% | $D \leq 3$ |
| | Hybrid Strategy (Ours) | **75.8** | **3.5%** | $D > 11$ |

**Key Advantages:** (1) **Type-specific adaptation**—our adaptive strategies automatically distinguish forgetting types and apply appropriate mitigation, while (Zheng et al., 2025)'s fixed freezing applies the same strategy regardless of forgetting type; (2) **Deep alignment promotion**—our strategies achieve $D > 10$ on average, while (Zheng et al., 2025)'s fixed freezing maintains shallow alignment ($D \leq 3$); (3) **Better performance**—our adaptive strategies outperform fixed freezing by 2.7-4.3% in accuracy and reduce forgetting rate by 4.5-5.6%; (4) **Real-time detection**—our framework enables early intervention, while (Zheng et al., 2025) relies on post-hoc analysis.

*Table 12.* Recovery Process Tracking

| Epoch | Alignment Score | Accuracy | Gradient Norm | Status |
|---|---|---|---|---|
| Initial (after forgetting) | 0.48 | 68.3% | 0.12 | Detected |
| After 0.5 epochs | 0.62 | 78.5% | 0.08 | Recovering |
| After 1.0 epochs | 0.78 | 88.2% | 0.05 | Recovering |
| After 1.2 epochs | 0.85 | 92.1% | 0.03 | Recovered |
| After 1.5 epochs | 0.87 | 94.3% | 0.02 | Stable |

**Recovery Process**   Recovery is rapid (1.2 epochs), confirming spurious forgetting can be quickly reversed.

*Table 13.* Computational Overhead Breakdown

| Component | Time Overhead | Memory Overhead | Frequency | Total Overhead |
|---|---|---|---|---|
| Alignment computation | +3% | +0.5GB | Every 100 steps | +5% |
| Reversibility analysis | +2% | +0.3GB | Every task | +3% |
| Dynamic tracking | +1% | +0.4GB | Every 50 steps | +2% |
| Adaptive freezing | +0.5% | +0.1GB | Every task | +1% |
| Selective repair | +0.5% | +0.2GB | When detected | +1% |
| **Total** | **+7%** | **+1.5GB** | | **+12%** |

**Computational Overhead**

**Hyperparameter Sensitivity**   We analyze sensitivity to key hyperparameters. Table 14 shows results.

*Table 14.* Hyperparameter Sensitivity Analysis

| Hyperparameter | Value | ACC | Identification Accuracy | Recovery Rate |
|---|---|---|---|---|
| $\tau_{\text{align}}$ | 0.6 | 74.2 | 82.1 | 88.3% |
| | 0.7 | 76.4 | 88.4 | 94.2% |
| | 0.8 | 75.8 | 85.3 | 91.5% |
| | 0.9 | 74.5 | 81.2 | 87.8% |
| $\tau_R$ | 0.5 | 75.1 | 84.2 | 90.1% |
| | 0.6 | 76.4 | 88.4 | 94.2% |
| | 0.7 | 75.9 | 86.3 | 92.5% |
| | 0.8 | 74.8 | 83.1 | 89.3% |
| Monitoring Frequency | 50 steps | 76.1 | 87.2 | 93.1% |
| | 100 steps | 76.4 | 88.4 | 94.2% |
| | 200 steps | 75.8 | 85.5 | 91.8% |

Optimal settings: $\tau_{\text{align}} = 0.7$, $\tau_R = 0.6$, monitoring every 100 steps.

